# Sequence Identification of Bioactive Peptides from Amaranth Seed Proteins (*Amaranthus hypochondriacus* spp.)

**DOI:** 10.3390/molecules24173033

**Published:** 2019-08-21

**Authors:** Alexis Ayala-Niño, Gabriela Mariana Rodríguez-Serrano, Luis Guillermo González-Olivares, Elizabeth Contreras-López, Patricia Regal-López, Alberto Cepeda-Saez

**Affiliations:** 1Chemistry Investigation Center, Universidad Autónoma del Estado de Hidalgo, Carretera Pachuca-Tulancingo km 4.5, Mineral de la Reforma Hidalgo C.P. 46067, Mexico; 2Biotechnology Department, Universidad Autónoma Metropolitana, Unidad Iztapalapa, Mexico City C.P. 55355, Mexico; 3Universidad de Santiago de Compostela, Campus Lugo, 15705 Santiago de Compostela, 27002 A Coruña, Spain

**Keywords:** amaranth protein, flavourzyme, alcalase, bioactive peptides, hydrolysates

## Abstract

*Amaranthus hypochondriacus* spp. is a commonly grown cereal in Latin America, known for its high protein content. The objective of this study was to separate and identify bioactive peptides found in amaranth seeds through enzymatically-assisted hydrolysis using alcalase and flavourzyme. Hydrolysis was carried out for each enzyme separately and compared to two-step continuous process where both enzymes were combined. The biological activity of the resulting three hydrolysates was analyzed, finding, in general, higher bioactive potential of the hydrolysate obtained in a continuous process (combined enzymes). Its fractions were separated by RP-HPLC, and their bioactivity was analyzed. In particular, two fractions showed the highest biological activity as ACE inhibitors with IC50 at 0.158 and 0.134, thrombin inhibitors with IC50 of 167 and 155, and antioxidants in ABTS assay with SC50 at 1.375 and 0.992 mg/L, respectively. Further sequence analysis of the bioactive peptides was carried out using MALDI-TOF, which identified amino acid chains that have not been reported as bioactive so far. Bibliographic survey allowed identification of similarities between peptides reported in amaranth and other proteins. In conclusion, amaranth proteins are a potential source of peptides with multifunctional activity.

## 1. Introduction

Many diseases that prevail nowadays could be tackled more efficiently, and even prevented, by combining a healthy diet with functional foods intake [1]. Among functional food components, we can find bioactive peptides, which are generally short sequences of amino acids encrypted in food proteins [2]. Bioactive peptides are released from proteins as a result of microorganisms’ metabolic activity during fermentation, by proteolytic enzymes, or finally by the action of gastrointestinal enzymes once proteins are ingested. Application of commercially available enzymes has become a simple and inexpensive way to access free amino acids and peptides from proteins [3].

Consequently, a vast number of studies using various proteinases target the preparation of hydrolysates from different protein sources, such as cereals and pseudocereals, in an ongoing effort to obtain highly active biopeptides [4,5,6,7]. One of the most important highly-consumed Mesoamerican original seeds, with a high content of proteins and an excellent amino acid balance, is amaranth. In Mexico, amaranth is consumed in a fresh form, offered by traditional confectionary and in typical dishes of Mexican cuisine [8]. Bioactive peptides found in amaranth exhibit various biological activities—such as anticholesterolemic, antihypertensive, antioxidant, and antithrombotic—and they are released mainly by in vitro digestion [9,10,11,12]. However, multiple bioactivities shown by the same peptide fraction of amaranth have never been studied.

In order to find peptide sequences with different biological activities, our research work focused on the separation and identification of the peptide sequences from amaranth proteins released during hydrolysis with two commercial enzymes, alcalase and flavourzyme, in separate and continuous hydrolysis processes. The multiple biological activities exhibited by peptides released from amaranth proteins through enzymatically-assisted hydrolysis have not been reported until now.

## 2. Results

### 2.1. Free Amine Groups Analysis during Enzymatic Hydrolysis

Once the hydrolitic enzymes were added to the protein sample, the progress of the reaction was monitored every 20 min by the measurement of free amine groups concentration released during the reaction course (Figure 1). It was observed that after 120 min of hydrolysis with alcalase (H1), the concentration of free amine groups equaled to 6170.53 ± 29.5 mg/L NH:^−^, whereas flavourzyme hydrolysis (H2) reached the highest concentration of free amine groups (5551.11 ± 33.83 mg/L NH:^−^) after 90 min of reaction. It was concluded that, in terms of enzymatic efficiency, alcasase overdoes flavourzyme in the release of peptide fractions. Following the analysis of H1 and H2, the combined two-step hydrolysis (H3) was carried out, and a final free amine groups concentration of 7468.89 ± 34.79 mg/L NH:^−^ was obtained after 40 min of reaction. This result proves that the continuous hydrolitic process (H3) was more efficient than using alcalase and flavourzyme separately.

### 2.2. Hydrolysates Bioactivity Analysis

Table 1 shows the assessment of the different biological activities exhibited by the hydrolysates H1, H2, and H3, determined during amaranth proteins hydrolysis. For thrombin inhibition essay, H1 hydrolysate showed the highest inhibitory potential (90%), followed by H2 hydrolysate which inhibited thrombin activity by approximately 80%. No significant difference was observed for H3 and H1 hydrolysates as thrombin inhibitors. When angiotensin-converting enzyme inhibition was evaluated, it was observed that H3 hydrolysate showed the highest inhibitory potential (58%), in comparison to H1 and H2 hydrolysates (49% and 39%, respectively). Finally, antioxidant activity was measured by three different methods based on the free radical scavenging principle, such as ABTS, DPPH, and FRAP. The highest antioxidant activity was observed by the H3 sample in DPPH and FRAP assays (388.94 µmol TE/100 g and 592.54 µmol Fe2 E/100 g, respectively), while H1 hydrolysate exhibited the highest antioxidant activity measured by the ABTS method (425.86 ± 0.66 mg TE/100 g). Noteworthy, the antioxidant potential of H3 hydrolysate increased fivefold in the DPPH assay and ninefold in the FRAP assay, compared to the control sample of unhydrolyzed amaranth proteins.

### 2.3. RP-HPLC Separation and Fraction Bioactivity

Since the H3 hydrolysate (obtained by application of continuous enzymatic hydrolysis) showed the highest bioactivity, it was chosen for further analysis in order to separate fractions and identify peptide sequences. In total, 56 fractions were obtained through RP-HPLC separation (see Appendix A), of which only 14 could be identified with known proteins. Moreover, protein content was determined in samples collected after 27 min (Table 2). In the case of the 14 fractions identified with protein, their bioactivity was evaluated by measurement of the IC50 in the ACE and thrombin inhibition assays, and SC50 in the ABTS assay. Results are gathered in Table 2.

For the antihypertensive potential, the IC50 calculated for the test fractions ranged from 0.134 to 0.808 mg/mL (Table 2). These values stay in agreement with the results reported elsewhere for amaranth protein hydrolysates [10,13]. On the other hand, values obtained in this work were lower than those found for other vegetable protein sources [14], or higher in comparison to buffalo milk [15]. Antioxidant potential based on the ABTS assay and expressed as SC50 varied from 0.992 to 6.931 mg/mL, and was slightly higher than that obtained for amaranth hydrolysates by in vitro digestion [16]. This may be explained by the fact that the type of bioactive peptides released depends not only on the protein nature, but also on the enzymes used in the particular case of hydrolysis [17,18]. Concerning the thrombin IC50 values, the results obtained were in the range of 0.992 to 38.46 mg/L, which stay in agreement with the values reported for amaranth hydrolyzed by alcalase and pepsin [12]. Finally, no statistically important differences were found for thrombin inhibition potential between H3 hydrolysate (alcalase and flavourzyme) investigated in this study, and amaranth hydrolyzed by alcalase and pepsin as reported in Sabbione et al. [12], which was also obtained via sequential two-enzyme process.

Based on the obtained results, fractions 22 and 45 were subjected to MALDI-TOF analysis (Figure 2), which allowed the identification of sequences with interesting multiple bioactivities. 

The structure of the aminoacidic chains and the corresponding protein source are presented in Table 3.

## 3. Discussion

Results obtained for the individual enzymatic hydrolysis were similar to those reported by Zhuang et al. [19] and Ma et al. [20], where it was observed that the enzymatic activity of alcalase and flavourzyme decreased after 90–120 min of the reaction course. The duration of enzymatic activity depends on the nature of the protein matrix and the amount of enzyme and substrate present in the medium. Additionally, other studies indicated a higher hydrolytic degree when only alcalase was used [21]. For H3 results, they are similar to those reported by Cumby et al. [22], who found that combined enzymatic hydrolysis increased peptide concentration when compared to hydrolysis performed solely with alcalase. It was also stated that hydrolysis with flavourzyme did not contribute to reach higher concentrations of free amine groups when alcalase was used in the first step of the combined enzymatic hydrolysis. On the other hand, the degree of hydrolysis depends on nature of protein and the specificity of the enzymes used [17,18].

Alcalase and flavourzyme have been used individually and in continuous hydrolytic processes applied to fragmentation of different food proteins in search of platelet inhibitory peptides [23,24]. In the case of amaranth proteins, the addition of a second enzyme different to flavourzyme, following hydrolysis with alcalase, has proven to enhance the inhibition potential of thrombin. This finding implies that the higher degree of hydrolysis, the greater bioactivity achieved [12,25]. In addition, Sabbione et al. [12] observed that the degree of hydrolysis of amaranth proteins, such as albumin and globulin, was an important factor in improved thrombin inhibition, reaching approximately 81%, which matches with the results obtained in present study (80% for H2, and 90% for H1 and H3).

The increase in thrombin inhibition activity may be due to the fact that amaranth protein has peptide sequences capable of inhibiting fibrinogen. These peptides are released by the action of proteolytic enzymes. Furthermore, it was observed that not only was the size of the peptide important for thrombin inhibition, but also the amino acid sequence, which might be homologous to the fibrinopeptides A and B from human fibrinogen (GGGVR-GP and PPSAR-GH, respectively) [26]. Therefore, antithrombotic activity is affected by the competition for platelet receptors between casoplatelin and the γ-chain of human fibrinogen (HHLGGAKQAGDV) [27,28], which is implicated in one of the three steps in thrombosis cascade reactions. Additionally, alcalase has been reported as an enzyme capable of releasing antithrombotic peptides form proteins like peanut [18], but no reports have been available so far on hydrolytic potential of flavourzyme to release bioactive peptides from amaranth.

In the case of antihypertensive activity, the results obtained in this study are similar to those reported by Ambigaipalan et al. [29]. They observed that the combination of two or more enzymes increased the inhibitory activity of angiotensin, converting the enzyme in date seed protein in a sequential enzymatic process. Many authors have reported that the variation in antihypertensive activity of the released peptides could be attributed to the differences in composition and hydrophobicity of the protein primary structure [30,31]. In this sense, hydrophobic residues of amino acids (leucine, valine, alanine, tryptophan, tyrosine, proline, and phenylalanine) bind at the ACE catalytic sites, acting as competitive inhibitors [32]. Nevertheless, milk proteins have been reported as a better source of bioactive peptides, especially those with antihypertensive activity reaching over 80% of ACE inhibition [33,34]. Furthermore, studies have been focused on peptides released from vegetable proteins, such as amaranth. In particular, 11S globulin was reported to show IC50 value in the ACE assay, ranging from 6.32 mM to 175 µM [9,35]. It has been known that the low molecular weight of peptides is a prerequisite for their antihypertensive activity [36]. In this study, peptides with the latter feature were identified using SDS-PAGE (data not shown), revealing typical weights of these biopeptides.

In addition to antihypertensive activity, peptide size (peptide chain length less than 20 amino acids) has been related to antioxidant capacity, showing that smaller peptides have greater potential [37]. This observation could explain the higher antioxidant activity obtained for H3 hydrolysate, which contained a higher concentration of free amine groups. In conclusion, during the course of enzymatic hydrolysis H1, H2, and H3 carried out in this study, peptides with different biological properties were released—for example, those able to chelate reactive agents, or donate electrons or hydrogen [38]. Moreover, in several studies in which sequential hydrolysis of proteins with different enzymes was performed, it has been observed that the antioxidant activity remained the same or might be increased by the addition of a second enzyme [39,40]. Thus, the application of two hydrolytic enzymes allows the release of new bioactive peptides, which could affect the antioxidant capacity of the hydrolysates. Consequently, it has been observed that protein hydrolysates might present higher antioxidant properties as they increase their content of small peptides [41].

For RP-HPLC peptide separation, the results obtained in this work are similar to those reported by Moronta et al. [42], who hydrolyzed amaranth proteins with alcalase. It was found that peptides with a higher content of polar amino acids showed higher anti-inflammatory activity. Nonetheless, a fraction found in the non-polar region of the chromatographic analysis (45 min) exhibited higher potential in terms of antihypertensive, antithrombotic, and antioxidant activity than other polar fractions. In some studies, it has been demonstrated that peptides containing hydrophobic amino acids might enhance biological capacities, such as antihypertensive and antioxidant activities [43].

As observed in Table, fraction 22 contained the longest peptide sequence, identified as ITASANEPDENKS, with a molecular weight of approximately 1.44 kDa, being the highest molecular weight found in both fractions. In the case of the peptide NIDMLRL, the last part of its sequence (-LRL) has been identified in silico as an inhibitor of ACE in amaranth. This asseveration may be explained by the presence of leucine as hydrophobic amino acid interacting with the active site of ACE [44].

Additionally, the LVRW sequence was found in fraction 22. The three amino acid chain LVR was previously described as a bioactive peptide with antihypertensive activity in fig sap having an IC50 lower than 20 µM [45]. Additionally, W might be also a bonding amino acid in the active site of ECA. On the other hand, the IC50 of ACE inhibitory potential found for fraction 45 was lower than for fraction 22. Fraction 45 contained VRWS, and the dipeptide VR was described as an antihypertensive agent by itself, with an IC50 of 52.80 µM [46]. In addition, even though it was bound to other amino acids, such as Y or SP, it retained its bioactivity [47,48]. Similarly, the presence of Tyr in the C-terminal end should promote binding to ACE and thus enhance its inhibition [49]. It was proved for the tripeptide IVY [50], also present in the sequence CIHNIVY in fraction 45. Finally, in this fraction, various peptides containing Tyr residue were found.

In previous works on antioxidant peptides, it was stated that peptides with a length between 5 and 16 amino acids showed antiradical activity [51]. The results obtained in the present study support these findings, since the potential of antioxidant peptides had a molecular weight in the range of 500–1400 Da and contained 4–13 amino acid residues (Table 3).

One of the peptides described in fraction 22 was LVRW, which could play a role as an antioxidant agent. Haung et al. [52] determined that the presence of the RW amino acid sequence in the C-terminal end of the polypeptide chain might be accounted for by its high antiradical activity. A similar effect was observed for the peptide ARVW, were the antioxidant activity of Trp is mainly due to its indole group [53].

In fraction 45, the DPKLTL sequence was identified, which might have antioxidant capacity owning to the presence of the DPK fragment (previously described as an antioxidant peptide). Its bioactivity could be explained by the presence of aspartic acid, which has the ability to donate electrons and hydrogen. On the other hand, this fragment is bound to hydrophobic amino acids, namely Pro and Leu, which in turn could enhance the radical scavenging abilities of this peptide [54]. Additionally, the DPKLTL peptide might exhibit antithrombotic activity due to the presence of three amino DPK acid residues, previously known for inactivating thrombin in its active site [25].

According to the work of Wang et al. [55], peptides with antithrombotic bioactivity usually contain 3–20 amino acid residues. Table 3 shows that the longest peptide identified in fraction 22 had a 13-amino acid chain. Moreover, it has been reported that sequences containing Val and Tyr might possess antithrombotic activity [23]. These amino acids were mainly found among peptide sequences present in fraction 45. On the other hand, Pro-Arg bonding N-terminal end of polypeptide chains has been reported as a thrombin inhibitor in its active site [26]. This bond is present in the structure of PRYDQY; however, more exhaustive studies are necessary on antithrombotic peptides as thrombin contain three main structural domains (a catalytic site and two exosites (I and II)), and enzyme inhibition can take place to different extents at any of these sites [26,56].

## 4. Materials and Methods

### 4.1. Sample and Treatments

Raw *Amaranth hypochondriacus* spp. seeds were obtained from Xochimilco, Mexico City, in July 2016. Seeds were ground in a Chopin mill, segregated according to their molecules size, and only a fraction with molecule size between 200 and 800 µm was selected for the experiments. Protein extraction was performed following the methodology described by Martínez and Añón [57] with some modifications. First, flour was defatted with n-hexane (10% *w*/*v*) for 24 h. Then it was suspended in deionized water 10% (*w*/*v*), and pH 9 was adjusted with NaOH. The crude was incubated for 30 min at room temperature and centrifuged during 20 min at 10,000 rpm. The soluble protein fraction found in the supernatant was precipitated by pH 5 adjustment, using HCl. The sample was centrifuged 20 min at 10,000 rpm, and the pellet pH was adjusted to 7. Following the lyophilization, the protein extract was stored at 4 °C until further use. This protein was called protein extract.

### 4.2. Enzymatic Hydrolysis

The enzymatic hydrolysis was performed according to the method described by Tironi and Añón [58]. Briefly, 5 g of the protein extract was diluted in 100 mL of deionized water (milli Q 18.2 MΩ*cm, Manufacturer, Darmstadt, Germany). For alcalase hydrolysis (H1), prior to the reaction, the pH of the solution was adjusted to 10, and alcalase (≥2.4 U/g, Anson Units; Sigma-Aldrich (St. Loui, MO, USA) was added at concentration of 8 µL/100 mg of the sample. For flavourzyme hydrolysis (H2), the pH of the reaction media was adjusted to 7 and flavourzyme (≥500 U/g; Sigma Aldrich, St. Louis, MO, USA) was added at concentration of 5 µL/100 mg of the sample. Finally, when both enzymes were used in a two-step continuous hydrolytic process (H3), after 2 h of hydrolysis with alcalase, the enzyme was inactivated by heating to 85 °C for 10 min. The solution was adjusted to pH 7 and hydrolysis with flavourzyme was carried out. For all three methods, hydrolysis was followed up during 4 h and progress of the reaction was monitored every 20 min. The aliquots were heated to 85 °C during 10 min and frozen until further use.

### 4.3. Free Amine Groups Analysis by TNBS Test

In order to determine free amine groups released during the enzymatic hydrolysis of the amaranth proteins, the 2,4,6-Trinitrobenzenesulfonic acid (TNBS) test was performed, with some modifications, according to the protocol described by Sashidhar et al. [59]. Briefly, 5% TNBS solution (Sigma-Aldrich, (St. Loui, MO, USA) was diluted in 0.21 M phosphate buffer (pH 8.2) to a final concentration of 1% (*v*/*v*). Two milliliters of the prepared substrate was added to 2 mL of phosphate buffer (0.21 M; pH 8.2), and 0.25 mL of the test sample. The mixture was incubated during 1 h at 50 °C and the reaction was stopped by the addition of 2 mL of 0.1 N HCl. The absorbance was read at 340 nm. Results were plotted against a calibration curve prepared by different glycine concentrations (0, 0.05. 0.1, 0.15, 0.2, and 0.25 mg/mL) using the following equation (R = 0.9972):
y = 0.004x + 0.134(1)

Results are expressed as milligrams of free amines per liter (mg/L NH:^−^).

### 4.4. Antihypertensive Activity

Inhibitory effect on angiotensin converting enzyme (ACE) was evaluated spectrophotometrically according to the method of Cushman et al. [60] using Hippuril-Histidyl-Leucine (St. Loui, MO, USA) as substrate. Briefly, 5 mM substrate solution was prepared in 0.1 M sodium borate buffer pH 8.3 containing 0.3 M sodium chloride. A 5 mM 100 µL aliquot of substrate solution was mixed with 40 µL of the test sample before adding 10 µL of angiotensin converting enzyme (EC 3.4.15.1, 5.1 U/mg; Sigma-Aldrich). The reaction mixture was incubated during 1 h 15 min at 37 °C and the enzyme was inactivated with 1 mL of 0.1 M HCl. The hippuric acid formed in this reaction was extracted with ethyl acetate, concentrated under reduced pressure, and finally re-dissolved in distilled water. The absorbance was measured at 220 nm in a GENESYS spectrometer.

The inhibitory activity of ACE was calculated using the formula:
% inhibitory activity = (AbsC − AbsM)/(AbsC −AbsB) × 100(2) where:

AbsC: Hippuric acid formed during the reaction with ACE without inhibitor.

AbsB: Hippuril-Histidyl-Leucine that did not react and was extracted with ethyl acetate.

AbsM: Hippuric acid formed after the reaction with ACE in the presence of inhibitory substance.

### 4.5. Antithrombotic Activity

To evaluate antithrombotic activity, the methodology developed by Zhang et al. [61] was followed, including modifications of sodium chloride concentration proposed by Pérez-Escalante et al. [56]. Absorbance at 405 nm was measured before adding the enzyme to a microplate reader (Power Wave XS UV-Biotek, software KC Junior, USA) and after 10 min of incubation with the enzyme at 37 °C. The percentage of inhibition (% inhibition) was calculated following the equation:
% inhibitory activity = [(C − CB) − (S − SB)]/(C − CB)*100(3) where:

CB (control blank): The initial absorbance of the negative control of inhibition.

C (control): The absorbance of the negative control after 10 min of incubation with thrombin.

SB (sample blank): The initial absorbance of the sample.

S (sample): The absorbance of the sample after 10 min of incubation with thrombin.

### 4.6. Antioxidant Activity

Antioxidant activity was evaluated by three different methods.

#### 4.6.1. ABTS test

Radical scavenging capacity was measured using the radical cation 2,2′-azino-bis 3-ethylbenzothiazoline-6-sulphonic acid (ABTS•+), which was produced by mixing 7 mM of ABTS•+ stock solution with 2.45 mM potassium persulfate in the dark at room temperature for 16 h prior to its usage. The ABTS•+ solution was diluted with deionized water, so that absorbance measured at 754 nm was of 0.70 ± 0.02. An aliquot of 20 μL of test sample was added to 980 μL of the diluted ABTS•+ solution, and after 7 min of incubation at room temperature, the absorbance readings were taken at 754 nm in a microplate reader (Power Wave XS UV-Biotek, soft-ware KC Junior, USA). Antioxidant capacity was expressed as milligrams Trolox equivalents per liter (mg TE/100 g) [62].

#### 4.6.2. DPPH Test

Radical scavenging activity was measured using 2,2-diphenyl-1-picrylhydrazyl (DPPH•) radical. An ethanolic solution (7.4 mg/100 mL) of the stable DPPH• radical was prepared. Then, 100 μL of the test sample was added to 500 μL of DPPH• solution, and it was left to sit at room temperature for 1 h. The solution was stirred and centrifuged at 3000 rpm for 10 min. Finally, absorbance of the supernatant was measured at 520 nm in a microplate reader. Antioxidant activity was expressed as micromole of Trolox equivalents per liter (TE μmol/100 g) [63].

#### 4.6.3. FRAP Test

FRAP antioxidant activity was evaluated according to the Benzie and Strein method [64], in which 0.3 M sodium acetate buffer pH 3.6, TPTZ, 20 mM FeCl3, and 5M FeSO4 were elaborated. For the preparation of FRAP buffer, TPTZ and FeCl_3_ were mixed at 10:1 ratio (*v*:*v*). Briefly, 30 µL of test sample was mixed with 900 µL of FRAP solution and 90 µL of distilled water, and agitated. The mixture was incubated during 10 min at 37 °C, after which the absorbance was measured at 593 nm in a microplate reader. Results were compared with a calibration curve constructed for FeSO_4_ standards at concentrations ranging from 0 to 1000 mM. Antioxidant activity was expressed as micromole equivalents of Fe (II) per 100 g (µmolEFeII/100 g).

### 4.7. Identification of Bioactive Peptides by RP-HPLC

#### 4.7.1. Sample Preparation

For sample preparation, 10 mg/mL of freeze-dried amaranth protein hydrolysates were prepared with phosphate buffer (pH 7.8). The crude mixture was stirred during 1 h at 37 °C and followed by centrifugation at 10,000 rpm for 10 min at room temperature. The resulting supernatant with corresponding soluble fractions was separated and stored.

#### 4.7.2. RP-HPLC Separation

Peptides were separated by reversed-phase chromatography on a HPLC (Waters, USA) system equipped with a C8 column (250 mm × 4.6 mm × 5 mm; Waters) and photodiode array detector (Spectra System Thermo Scientific, USA) A gradient elution was applied from 100 to 0% A in 56 min at flow rate of 1 mL/min. A binary mobile phase consisted of solvent A: 0.065% trifluoroacetic acid [TFA] in water/acetonitrile [ACN] 98:2, and solvent B: 0.065% TFA in water/ACN 35:65. Detection was performed at 280 nm and temperature was set at 40 °C. The injection volume was 200 µL. Fractions were manually collected every 1 min in Eppendorf tubes and protein content was evaluated by the Bradford assay. Finally, samples were lyophilized (Labconco DrySystem/freezone 4.5) and bioactivity analysis was carried out only on the lyophilized samples containing protein.

#### 4.7.3. MALDI-TOF Spectrometry

The collected fractions with the highest bioactivities were filtered through a Minisart RC4 filter (0.45 µm) and analyzed by a matrix-assisted laser desorption ionization (MALDI) mass spectrometer, equipped with a delayed extraction source and a 355 nm pulsed nitrogen laser. A MALDI scoutMTPTM was run in the linear mode. A 100-times diluted sample was mixed with 1 volume of 20 mg/mL of sinapinic acid in acetonitrile/water 50:50 (*v*/*v*). Finally, 0.5 µL of the mixture was deposited onto the MALDI target plate. All spectra were the result of signal averaging of 200 shots. The MALDI-TOF/TOF MS/MS was run in the positive refractor mode. The peptide sequencing was performed by processing the MS/MS spectra using Auto eXecute software. Mascot (Matrix Science Inc., Boston, MA, US) software was used to identify and characterize peptide structures.

### 4.8. Statistical Analysis

All data were obtained in triplicate and expressed as mean ± standard deviation (SD). The data were analyzed by one-way analysis of variance (ANOVA) tests, and the differences among means were compared using the Tukey test with significance level set at *p* < 0.05, using the SPSS^®^ System for WINTM version 15.0 (IBM® Armonk, New York, NY, USA).

## 5. Conclusions

Amaranth protein hydrolysate, obtained through enzymatic reaction with alcalase and flavourzyme in a sequential two-step hydrolytic process, may be a source of bioactive peptides. Sequences were different from those obtained in more commonly performed enzymatic hydrolysis using pepsin and pancreatin for amaranth proteins digestion. The biological activities identified for some peptide sequences in this study have been proven in other food sources. Moreover, novel amino acid chains with possible multi-functional activities were identified. This is the first report of the multiple bioactivities of peptide fractions derived from hydrolyzed amaranth proteins. Additionally, this study shows that amaranth hydrolyzed with alcalase and flavourzyme could be used in the nutraceutical industry, as a value-added ingredient with multi-functional bioactive properties.

## Figures and Tables

**Figure 1 molecules-24-03033-f001:**
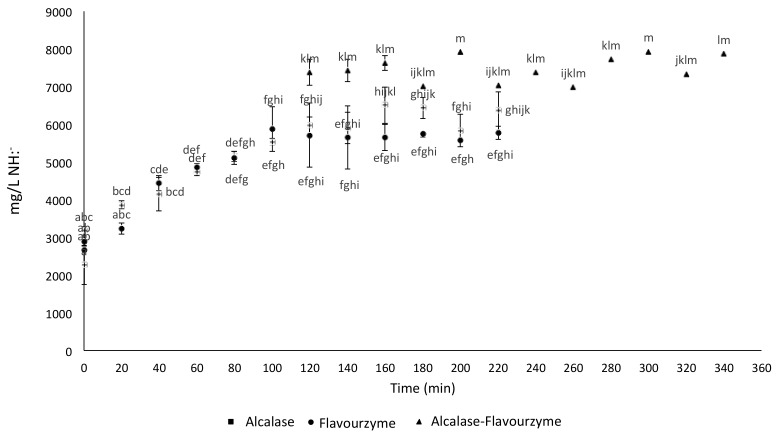
Free amine groups concentration during amaranth protein hydrolysis. * Different letters indicate statistically significant differences between treatments.

**Figure 2 molecules-24-03033-f002:**
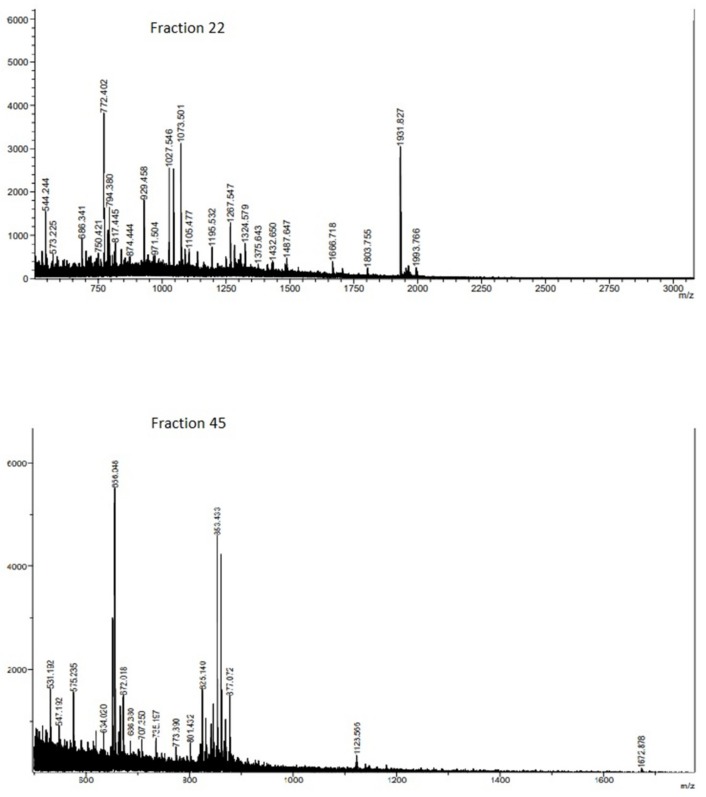
Fraction 22 and 45 MALDI-TOF mass Spectra.

**Table 1 molecules-24-03033-t001:** Bioactivity potential of amaranth protein hydrolysates after treatment with alcalase (H1), flavourzyme (H2), and in two-step combined hydrolysis (H3).

Hydrolysis	ACE Inhibition (%)	Thrombin Inhibition (%)	Antioxidant Activity		
			DPPH (µmol Trolox E/100 g)	ABTS (mg Trolox E/100 g)	FRAP (µmol Fe2 E/100 g)
**Amaranth Protein**	10.58 ± 1.19 ^d^	11.90 ± 10.10 ^c^	76.66 ± 1.60 ^d^	115.65 ± 10.30 ^d^	63.37 ± 5.72 ^c^
**H1**	49.49 ± 1.47 ^b^	92.85 ± 3.36 ^a^	340.17 ± 10.95 ^b^	425.86 ± 0.66 ^a^	241.70 ± 9.38 ^b^
**H2**	39.77 ± 2.15 ^c^	80.95 ± 13.46 ^b^	274.03 ± 10.84 ^c^	398.36 ± 3.62 ^c^	226.29 ± 11.20 ^b^
**H3**	58.53 ± 2.58 ^a^	92.85 ± 3.36 ^a^	388.94 ± 2.73 ^a^	404.90 ± 1.52 ^b^	592.54 ± 29.29 ^a^

Values are shown as mean ± standard deviation (*n* = 3); values in the same column with different superscript letters are significantly different (*p* < 0.05).

**Table 2 molecules-24-03033-t002:** Bioactivities of amaranth protein fractions (Angiotensin I-Converting Enzyme and Thrombin inhibitory activity (IC_50_, mg/L)) and antioxidant activity (ABTS radical scavenging; SC_50_, mg/L).

Fraction	ACE (IC_50_)	Thrombin (IC_50_)	ABTS (SC_50_)	Peptide Concentration (mg/L)
2	0.332 ^cd^	38.46 ^i^	4.204 ^e^	0.2125
3	0.442 ^e^	4.36 ^h^	NI	0.8375
9	NI	0.426 ^f^	NI	0.9062
18	0.614 ^f^	2.65 ^g^	2.538 ^d^	0.7125
19	0.173 ^b^	0.183 ^b^	NI	0.3750
22	0.158 ^ab^	0.167 ^ab^	1.375 ^b^	0.4687
23	NI	0.349 ^e^	2.809 ^d^	0.5625
27	0.808 ^c^	0.402 ^f^	1.616 ^c^	0.3125
28	0.346 ^d^	0.135 ^a^	1.728 ^c^	0.0937
32	0.192 ^b^	0.298 ^d^	6.931 ^g^	0.4687
34	0.317 ^cd^	0.247 ^c^	2.593 ^d^	0.4937
39	NI	0.247 ^c^	5.561 ^f^	0.4375
40	0.298 ^c^	0.26 ^cd^	4.547 ^e^	0.8375
45	0.134 ^a^	0.155 ^a^	0.992 ^a^	0.8125

Values for ACE, thrombin, and ABTS (mg peptide/mL) are mean ± SD (*n* = 3); values in the same column with different superscript letters are significantly different (*p* < 0.05). NI: not identified.

**Table 3 molecules-24-03033-t003:** Peptides identified in the most active fractions from amaranth hydrolysate and their corresponding protein source.

Fraction	Mass *m*/*z*	Calc MH^+^	Sequence	Protein
Fraction 22				
**34**	1375.6435	1375.6341	ITASANEPDENKS	Agglutinin
**3**	573.2252	573.3516	LVRW	Agglutinin
**16**	874.4448	874.4813	NIDMLRL	Granule bound starch synthase I
**12**	794.3805	794.4203	RPVFEF	Granule bound starch synthase I
**5**	686.3414	686.4081	DPKLTL	Granule bound starch synthase I
**3**	573.2251	573.3617	IKEAL	Granule bound starch synthase I
**13**	812.3607	812.4265	NVEVHKS	Cystatin
**Fraction 45**				
**27**	853.4330	853.4329	HVQLGHY	Agglutinin
**14**	707.3505	707.3212	SQIDTGS	Agglutinin
**14**	707.3502	707.3185	NWACTL	Agglutinin
**4**	547.1921	547.2997	VRWS	Agglutinin
**29**	861.3847	861.4299	CIHNIVY	Granule bound starch synthase I
**26**	845.4098	845.4254	EGTESIPL	Granule bound starch synthase I
**24**	841.4242	841.3841	PRYDQY	Granule bound starch synthase I
**19**	823.4281	823.3696	MSNIDML	Granule bound starch synthase I
**13**	686.3805	686.4080	DPKLTL	Granule bound starch synthase I
**6**	619.2805	619.3566	IPSRF	Granule bound starch synthase I
**3**	531.1927	531.3042	ARVW	Granule bound starch synthase I
**2**	505.1907	505.2447	CQAAL	Granule bound starch synthase I
**1**	503.1730	503.2715	EELL	Granule bound starch synthase I
**1**	503.1731	503.2823	LGVAGS	Granule bound starch synthase I

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
