# Peer review of "Sequence Identification of Bioactive Peptides from Amaranth Seed Proteins (Amaranthus hypochondriacus spp.)"

_molecules, 2019, doi:10.3390/molecules24173033_

Round 1
Reviewer 1 Report
Manuscript: ISSN 1420-3049
In the present manuscript, Sequences identification of bioactive peptides derived of proteins hydrolyzed from amaranth (Amaranthus hypochondriacus) by enzymatic process is reported is presented.
The interest for generating, isolating, and identifying new source of potential bioactive peptides in nowadays of great interest and is the new direction for preventing many diseases, such as cardiovascular one.
After my revision I suggest the manuscript, before being published, necessarily requires an extensive revision of the English language.
Some other point concerning methodology and form should be addressed by several modifications.
Materials and methods section:
Line 209: What is semolina fraction? If you referred to molecular sieve, is preferable to use the correct name.
Line 302: Report the % of TFA in CAN. Is not necessary report the ml used for acidifying the LC phases.
Line 305: Is Bradford suitable for quantifying peptides? I suggest using more appropriate spectrophotometric assay, such as OPA. Please, modify this part.
Line 315: Explain more in details how the identification of peptide sequences was performed. Which kind of proteins database, i.e. SwissProt or NCBInr was used for identifying peptides.
In the conclusion part
Line 324: “…. showing that this hydrolysis method may release different sequences that those in in vitro digestion, which is the most common method for the release of bioactive peptides from amaranth protein.”. What the authors do mean? The hydrolysis that authors performed is already an in vitro digestion. Maybe they want to write in vivo??? Please modify this part.
Line 325-328: Please rearrange the all sentences. Is makes confusion.
Results
2.1 Section: Enzymatic Hydrolysis: I suggest presenting the data by Figure. For the reader it is easier to follow the data, presenting a trend over time.
Line 78: “…..H3 was the method with higher results” Could the author revise the English form? i.e. The hydrolysates obtained by applying the H3 hydrolysis… or something like that is preferable.
Line 91: “It got values”…reply with: results indicate…….
Table 3: Tentatively?????? Is not acceptable publishing something of which the authors do not declare certainty.
Discussion:
Line 116: “They mentioned that hydrolysis with alcalase does not allow the release of specific substrates for flavourzyme.” Revise this sentence: Two enzymes have different proteolytic activity. Or….amaranth proteins are differently hydrolysed by alcalase or flavoryme.
Line 119: continuos and separated process. What the author means? Explain it better.
Line 149: Maybe I did not understand well. The authors performed SDS-PAGE to visualize the peptides? In general is used a Tricine SDS-PAGE for peptide visualization.
Line 46: “ … and identify these chains of peptides….” Replace with: peptide sequences.
Author Response
English version of the article was reviewed and it was corrected
MATERIAL AND METHODS
Line 209: What is semolina fraction? If you referred to molecular sieve, is preferable to use the correct name.
Molecular size was used instead semolina
Line 302: Report the % of TFA in CAN. Is not necessary report the ml used for acidifying the LC phases.
Concentration in percentage was added.
Line 305: Is Bradford suitable for quantifying peptides? I suggest using more appropriate spectrophotometric assay, such as OPA. Please, modify this part.
In order to analyze the free amino groups concentration, TNBS method was used. Bradford method was used to standardize the protein content in each sample before the electrophoresis analysis.
Line 315: Explain more in details how the identification of peptide sequences was performed. Which kind of proteins database, i.e. SwissProt or NCBInr was used for identifying peptides.
The database used was added at material and method section. Mascot (Matrix Science Inc., Boston, MA, US) software was used to identify and characterize peptide structures.
CONCLUSION
Line 324: “…. showing that this hydrolysis method may release different sequences that those in in vitro digestion, which is the most common method for the release of bioactive peptides from amaranth protein.”. What the authors do mean? The hydrolysis that authors performed is already an in vitro digestion. Maybe they want to write in vivo??? Please modify this part.
The name of each enzyme used was added at the conclusion section, according the explanation in material and methods.
Line 325-328: Please rearrange the all sentences. Is makes confusion.
Sentences were rearranged.
RESULTS
2.1 Section: Enzymatic Hydrolysis: I suggest presenting the data by Figure. For the reader it is easier to follow the data, presenting a trend over time.
Figure identify as figure 1 was added
Line 78: “…..H3 was the method with higher results” Could the author revise the English form? i.e. The hydrolysates obtained by applying the H3 hydrolysis… or something like that is preferable.
Redaction was changed
Line 91: “It got values”…reply with: results indicate…….
Redaction was changed
Table 3: Tentatively?????? Is not acceptable publishing something of which the authors do not declare certainty.
The name of the table was changed and the word tentatively has been erased
DISCUSSION
Line 116: “They mentioned that hydrolysis with alcalase does not allow the release of specific substrates for flavourzyme.” Revise this sentence: Two enzymes have different proteolytic activity. Or….amaranth proteins are differently hydrolysed by alcalase or flavoryme.
Redaction was changed in order to make it understandable
Line 119: continuos and separated process. What the author means? Explain it better.
The sentences was rearranged for explaining better the action of the enzymes (... flavourzyme is not a factor to increase the concentration of free amino groups when alcalase is used as a previous hydrolytic enzime. )
Line 149: Maybe I did not understand well. The authors performed SDS-PAGE to visualize the peptides? In general is used a Tricine SDS-PAGE for peptide visualization.
The objective of using the SDS-PAGE technique was to observe peptide fractions less than 6 kDa. This is because these molecular weights have been reported for peptides with bioactive action. In fact, this analysis was used to observe the increase of peptide concentration and make a relation between this increasing and the bioactivity found. Actually, the tricine technique is generally used when it wants to separate peptide fractions between 2 and 6 kda. This is due to the resolution of the technique.In our case, separation and identification peptides was performed by HPLC and MALDI.
Line 46: “ … and identify these chains of peptides….” Replace with: peptide sequences.
Redaction was changed.
Reviewer 2 Report
Comments on the manuscript
Manuscript ID: molecules-529723
Recommendation: Reject and resubmit
The manuscript: “Sequences identification of bioactive peptides derived of proteins hydrolyzed from amaranth (Amaranthus hypochondriacus) by enzymatic process” had the goal to separate and identify bioactive peptides from Amaranthus hypochondriacus proteins, and using different assays to evaluate their anti-oxidant properties and thrombin and angiotensin-converting enzyme inhibition activities.
Although the interesting subject, in present form the manuscript is unacceptable for publishing in this Journal, having the serious omissions that have to be corrected.
English should be corrected, and before resubmitting, authors are advised to check the text with English native speakers (some of the places where the changes in English are necessary: lines 21,22; lines 68-71; lines 82, 83; lines 88-94, lines 143-144, ……).
The comments:
Authors should provide the full Latin name of all mentioned species in the manuscript (Latin name, author’s name, family name). Afterwards, throughout the text the shorten forms or common name should be used.
Abstract
The term “sequence process” mentioned in Abstract is unclear (lines 19,20).
Please, use the appropriate terms to define the performed experiments.
Introduction
Authors are asked to provide the explanation why specifically alcalase and flavourzyme were chosen for conducting the hydrolysis?
Line 40 - correct “Amaranth” into “amaranth”
What was the hypothesis to set the experiment for hydrolysis first treating the investigated samples separately with the mentioned enzymes, and then performing the hydrolysis by using one enzyme, and subsequently the other one?
Please, make sure that the NOVELTY or new aspect of the study, which has not been reported so far, has been stressed.
Results:
Please, make sure, when first mentioning the abbreviations, they were explained (H1, H2, H3)
Please, give the explanation why only ABTS test was used in further evaluation of anti-oxidant activity of the chosen fractions shown to be the richest in protein content.
Line 88, “in vitro” should be in italic
Table 1 - amaranths proteins (or amaranth protein fractions)
There is no Figure 1 attached to the submitted paper.
The novelty of amino-acids chains identified in fractions 22 and 45 was not confirmed
Table 3 - authors did not provide the format of sequence
Table 3 - legend is incomplete - there is no explanation what “peak” means…
Please, correct “aglutinina”
What does “granule boung starch synthase I” mean?
Discussion:
Please, use Past Tense when discussing the obtained results
Line 115 - please, give the explanation what “…sequence hydrolysis” means, especially as the term “peptide sequence”, “…to identify possible sequence…” were used in quite different contest.
Lines 127-128, please make the statement clear
Lines 133, 134 - please make the statement clear
Please, give the explanation and scientific clarification what was the reason to draw the conclusion that the observed “thrombin inhibition activity” might be consequence of the specific peptide sequence capable to inhibit fibrinogen, mentioning that “…the peptidic sequences…” “…may be homologous in position to the γ-chain sequence of human fibrinogen.” - please give the specific peptide sequence that the abovementioned statement referred to.
Please, explain the statement “The addition of a second enzyme allows the release of new bioactive peptides and the sequence released affect the antioxidant capacity”. Please, give the scientific ground of used combination of the alcalase and flavourzyme to allow the release of new bioactive peptides.
Line 160: Define “the most polar peptides”.
Materials and methods:
Please, give the sufficient explanation why for experiment with alcalase pH of 10 was chosen, as the recommend pH for it was 7-8.5
Where were the results of proteolytic capacity presented?
Why was DPPH solution prepared in almost two fold greater concentration then it was recommended for performing DPPH test?
Please, correct 5mm into 5microm (line 300).
Why was the analytical column used for preparative analysis?
How was the lyophilization of the collected samples performed?
Conclusion
Please, specify those amino acids chain for which it was stated that are new.
Author Response
COMMENTS
Authors should provide the full Latin name of all mentioned species in the manuscript (Latin name, author’s name, family name). Afterwards, throughout the text the shorten forms or common name should be used.
It has been reviewed all the text and the name of mentioned species are provide in Latin name. Afterwards, common name were used.
English should be corrected, and before resubmitting, authors are advised to check the text with English native speakers (some of the places where the changes in English are necessary: lines 21,22; lines 68-71; lines 82, 83; lines 88-94, lines 143-144, ……).
The english version of the article was reviewed and it has been corrected
ABSTRACT
The term “sequence process” mentioned in Abstract is unclear (lines 19,20). Please, use the appropriate terms to define the performed experiments.
The term "sequence process" was changed by reaction in one step and combined reaction.
INTRODUCTION
Authors are asked to provide the explanation why specifically alcalase and flavourzyme were chosen for conducting the hydrolysis?
Alcalase is a highly effective bacterial endoprotease, developed especially for the hydrolysis of all types of proteins.
Flavourzyme is a protease/peptidase complex of fungal origin, developed for the hydrolysis of proteins under neutral or slightly acidic.
Both enzymes, have the purity specifications recommended by food grade enzymes established by de Board of the FAO/WHO Expert Committee JECFA and FCC.
Line 40 - correct “Amaranth” into “amaranth”
It was changed
What was the hypothesis to set the experiment for hydrolysis first treating the investigated samples separately with the mentioned enzymes, and then performing the hydrolysis by using one enzyme, and subsequently the other one?
The specificity of enzymes could provide different peptide sequences after the hydrolysis. It could take advantage of that in order to search highest bioactivity in the peptide fractions. Many times when a combined hydrolytic reaction is carried out the bioactivity of the fractions increase, because the specific amino-acidic chains. in another hand, many reports about bioactive peptides mention that, the bioactivity is related in both, the specific aminoacid and the size of the peptide chain.
Please, make sure that the NOVELTY or new aspect of the study, which has not been reported so far, has been stressed.
The new aspect of the research was added in the introduction and in the conclusion section.
RESULTS
Please, make sure, when first mentioning the abbreviations, they were explained (H1, H2, H3)
H1, H2 and H3 were defined at first in the results section
Please, give the explanation why only ABTS test was used in further evaluation of anti-oxidant activity of the chosen fractions shown to be the richest in protein content.
In other studies, only one known radical is used to know SC50 of proteinic fractions, this due the lack in the amount of protein of each fraction.
Line 88, “in vitro” should be in italic
It was changed
Table 1 - amaranths proteins (or amaranth protein fractions)
The name of the table was changed
There is no Figure 1 attached to the submitted paper.
Figure 1 and 2 were attached
The novelty of amino-acids chains identified in fractions 22 and 45 was not confirmed
The novelty was not confirmed, however, many of the aminoacidic chains are not reported before this study. In order to be more specific, an explanation was written.
Table 3 - authors did not provide the format of sequence
The format to describe the sequences is FASTA format
Table 3 - legend is incomplete - there is no explanation what “peak” means…Please, correct “aglutinina”
"Protein fraction" was used instead "peak". "Aglutinina" was changed by "agglutinin"
What does “granule boung starch synthase I” mean?
It was changed to Granule Bound Starch Synthase
DISCUSSION
Please, use Past Tense when discussing the obtained results
Tenses were changed using past.
Line 115 - please, give the explanation what “…sequence hydrolysis” means, especially as the term “peptide sequence”, “…to identify possible sequence…” were used in quite different contest.
"Sequence hydrolysis" was changed by "peptide". "Peptide sequence size" was changed by "peptide size". "...to identify possible sequence" was eliminated when the sentence was rearranged.
Lines 127-128, please make the statement clear
The statement was changed and is clearer
Lines 133, 134 - please make the statement clear
The statement was changed and is clearer
Please, give the explanation and scientific clarification what was the reason to draw the conclusion that the observed “thrombin inhibition activity” might be consequence of the specific peptide sequence capable to inhibit fibrinogen, mentioning that “…the peptidic sequences…” “…may be homologous in position to the γ-chain sequence of human fibrinogen.” - please give the specific peptide sequence that the above mentioned statement referred to.
The sequences were added and the discussion was changed according the reviewer recommendation
Please, explain the statement “The addition of a second enzyme allows the release of new bioactive peptides and the sequence released affect the antioxidant capacity”. Please, give the scientific ground of used combination of the alcalase and flavourzyme to allow the release of new bioactive peptides.
An explanation was added
Line 160: Define “the most polar peptides”.
The polar nature of amino acids in peptides was added in the statement.
MATERIALS AND METHODS
Please, give the sufficient explanation why for experiment with alcalase pH of 10 was chosen, as the recommend pH for it was 7-8.5
Where were the results of proteolytic capacity presented?
"Free amine groups analysis" was used instead "proteolytic capacity". It was changed in both, material and methods and results section.
Why was DPPH solution prepared in almost two fold greater concentration then it was recommended for performing DPPH test?
It has been observed, concentrations near to 0.1 mM of DPPH radical has better resolution when the technic is performed. In our research group we always used that concentration. If is a recommendation of the reviewer, we could specify this situation in the text.
Please, correct 5mm into 5microm (line 300).
It was changed
Why was the analytical column used for preparative analysis?
The recovery volume after the HPLC separation with the analytical column, was enough in order to develop the identification of peptides during MALDI-TOF analysis. In addition, the volume of the enzymatic reaction were not enough for using a preparative column.
How was the lyophilization of the collected samples performed?
It was added lyophilization conditions
Conclusion
Please, specify those amino acids chain for which it was stated that are new.
A new sentence was added in the conclusion in order to specify the novelty of the study specially in those aminoacid chains that it has been never reported.

Reviewer 3 Report
In the submitted manuscript, the reversed-phase LC coupled with mass spectrometry was used for the identification of peptides enzymatically released from amaranth. The authors are presenting method and the results, which in my opinion requires further improvement before accepting for publication. Bellow, I am emphasizing the specific comments:
Abstract - first sentence should be omitted.
Introduction, line 41 - proper preposition should be used instead of "into".
Results, line 51 - free amino groups were evaluated - consider rephrasing (were determined? identified?). Line 78-79 the sentence should be rephrased - it is not clear.
Table 3 and connected results - authors have used high-resolution mass spectrometry, however the m/z values calculated and identified are shown only in three decimals. Please provide four decimals and accuracy between calculated and observed m/z values. This will strengthen the presented results.
Experimental part, line 300 - brand of the column is not given. Injection volume is missing.
Line 307 - title of subsection is misspelled. Line 311 - what does it mean 1 volume of matrix solution? More specific information about MS experiments are needed.
Author Response
English version of the article was reviewed and it was corrected
ABSTRACT
first sentence should be omitted.
It was erased as the reviewer recommended
Introduction, line 41 - proper preposition should be used instead of "into".
"in" was used instead "into"
Results, line 51 - free amino groups were evaluated - consider rephrasing (were determined? identified?).
"determined" was used instead "evaluated"
Line 78-79 the sentence should be rephrased - it is not clear.
Redaction was changed
Table 3 and connected results - authors have used high-resolution mass spectrometry, however the m/z values calculated and identified are shown only in three decimals. Please provide four decimals and accuracy between calculated and observed m/z values. This will strengthen the presented results.
Four decimals were added to the results
Experimental part, line 300 - brand of the column is not given. Injection volume is missing.
The column brand and the injected volume was added
Line 307 - title of subsection is misspelled.
"RP-HPLC separation" was written as subsection and "Spectometry" was well written
Line 311 - what does it mean 1 volume of matrix solution? More specific information about MS experiments are needed.
"Matrix solution" was erased and the specification of the solution was rearranged.

Round 2
Reviewer 1 Report
Thank you for the provided revisions. I considered the manuscript as publishable.
Author Response
thanks for trusting our work
Reviewer 2 Report
Comments on the manuscript
Manuscript ID: molecules-529723
Recommendation: Accept with major changes
The authors provided some of the requested changes in the submitted revision of the of manuscript “Sequence identification of bioactive peptides from amaranth seed proteins (Amaranthus hypochondriacus)”.
Still some shortcomings remained:
Figures 1 and 2 are the same in the supplementary materials, they have the same name. Figure 3 submitted as supplementary material is the same as Figure 1 in the submitted version of the manuscript – is this supplementary material, causing only the confusion, without any explanation, necessary?
During the revision process, authors made numerous typo and other mistakes – the words reminded together, or there is an extra space between the words. Authors should pay attention and check thoroughly the text before resubmitting. Still, English needs improvement (lines 185, 223, 231, 243…..)
Authors did not provide the Family of the investigated species. In addition, they used the commercial source, without valid identification of the plant material used in the experiment and prior to the analysis. Please, give the voucher number of the herbal specimen and the information who performed the identification of the seeds used in the experiments.
In addition, it is not explained why amaranth was used in the investigation (besides the statement: “One of the most important Mesoamerican origin seeds with a high content of proteins and an excellent amino acid balance is amaranth. In Mexico, amaranth is consumed in a fresh form offered by traditional confectionary and in typical dishes of Mexican cuisine” ….. Introduction still lacks the justification of the chosen subject of the investigation. The novelty of this work has still not been stressed.
Author Response
Figures 1 and 2 are the same in the supplementary materials, they have the same name. Figure 3 submitted as supplementary material is the same as Figure 1 in the submitted version of the manuscript – is this supplementary material, causing only the confusion, without any explanation, necessary?
It was a mistake. The correct figures has been submitted
During the revision process, authors made numerous typo and other mistakes – the words reminded together, or there is an extra space between the words. Authors should pay attention and check thoroughly the text before resubmitting. Still, English needs improvement (lines 185, 223, 231, 243…..)
The manuscript has been checked and it was submitted the correct version. English was revised too.
Authors did not provide the Family of the investigated species. In addition, they used the commercial source, without valid identification of the plant material used in the experiment and prior to the analysis. Please, give the voucher number of the herbal specimen and the information who performed the identification of the seeds used in the experiments.
Family of Amaranth was not investigated, because it was not the aim of the work. However, the name was completed whit the specification ssp in all the text.
In addition, it is not explained why amaranth was used in the investigation (besides the statement: “One of the most important Mesoamerican origin seeds with a high content of proteins and an excellent amino acid balance is amaranth. In Mexico, amaranth is consumed in a fresh form offered by traditional confectionary and in typical dishes of Mexican cuisine” ….. Introduction still lacks the justification of the chosen subject of the investigation. The novelty of this work has still not been stressed.
The explanation of the importance of amaranth, was added.
The justification is in relation with the importance, but a new phrase was added after the justification in introduction.
Part of the justification is related with the novelty and it was added in the text.
Reviewer 3 Report
The revised version of the manuscript is considerably improved with respect to the previous version. I would suggest accepting the manuscript for the publication in the present form.
Author Response
thanks for trusting our work